# CURIE: TOWARD RIGOROUS AND AUTOMATED COMPUTER SCIENCE EXPERIMENTATION WITH AI AGENTS

## ABSTRACT

Scientific experimentation demands *rigor* in reliability, methodical control, and interpretability to yield meaningful results. Despite the growing capabilities of large language models (LLMs) in automating different aspects of the scientific process, automating rigorous experimentation remains a significant challenge. To address this gap, we propose `Curie`[1], an AI agent framework designed to embed rigor into the experimentation process through three key components: an intra-agent rigor module to enhance reliability, an inter-agent rigor module to maintain methodical control, and an experiment knowledge module to enhance interpretability. To evaluate `Curie`, we design a novel experimental benchmark composed of 46 questions across four computer science domains, derived from influential research papers, and widely adopted open-source projects. Compared to the strongest baseline tested, we achieve a 3.4× improvement in correctly answering experimental questions. `Curie` is open-sourced at https://anonymous.4open.science/r/Curie-689B/.

## 1 INTRODUCTION

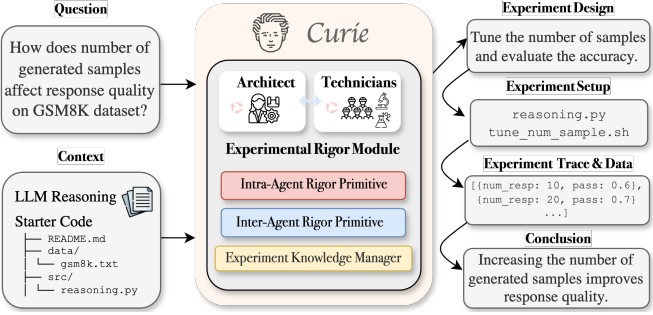

Figure 1: Curie overview.

Scientific research drives AI progress, advancing the development of the computer science discipline. At the heart of this endeavor lies experimentation—a disciplined intellectual pursuit that transforms human curiosity, expressed through bold hypotheses, into verifiable knowledge. Experimentation thrives on creativity, as new ideas fuel discovery. Yet it also depends on rigor—ensuring that research is methodologically sound and its findings are trustworthy (Armour et al., 2009; Gill & Gill, 2020).

In recent years, many works (Zhang et al., 2024b; Kramer et al., 2023; Lu et al., 2024a) leveraging large language models (LLMs) to automate scientific research have emerged (§2.3). These solutions typically rely on ad-hoc prompt-based methods to mimic scientific workflows, prone to hallucination. While effective for creative tasks such as literature review and brainstorming, these approaches remain limited in their ability to support rigorous experimentation, a largely unexplored capability.

---

[1] **Name disambiguation.** The name *Curie* was used by (Cui et al., 2025) for evaluating LLMs on scientific reasoning. In contrast, our work focuses on scientific experimentation, a substantially different problem domain.

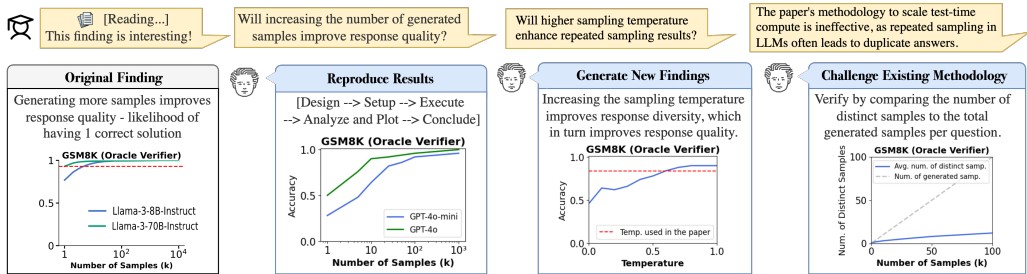

Figure 2: Case Study. `Curie` can help researchers validate, expand, and critique existing research on the benefits of repeated sampling in LLM reasoning (Brown et al., 2024). The first panel (Original Finding) presents a result from the original paper. `Curie` confirms this finding through rigorous experimentation in the second panel (Reproduce). The third panel (Extend) has `Curie` exploring the impact of sampling temperature on repeated sampling. The final panel (Challenge) shows `Curie` identifying a limitation in the original methodology, suggesting an avenue for future research.

More specifically, rigorous experimentation (§2.2) involves a *methodical procedure* that includes formulating hypotheses, designing experiments, executing controlled trials, and analyzing results. Achieving *reliability* at every step is essential to ensure that the results are accurate, reproducible, and scientifically meaningful. Finally, all procedures and results must be documented in a well-structured and *interpretable* manner, facilitating verification and collaboration across the community.

To meet these requirements, we propose `Curie`, an AI agent framework representing the first step toward rigorous and automated experimentation (§3). As shown in Fig. 1, `Curie` takes an experimental question and relevant context (e.g., domain-specific knowledge or starter code) as input. The Architect Agent generates high-level experimental plans, coordinates the process, and reflects on findings to guide subsequent steps. Working in unison, our Technician Agents focus on carefully implementing and executing controlled experiments following these plans.

At the core of `Curie`, the **Experimental Rigor Engine** preserves agent creativity while embedding rigor throughout the experimentation process. This is achieved via three key modules: (1) The *Intra-Agent Rigor Module* safeguards *reliability* of individual agents with a set of extensible rigor policies (e.g., validating that experiments align with objectives and setups are reproducible). (2) The *Inter-Agent Rigor Module* maintains methodical control over agent coordination, ensuring correct task transitions and efficient task scheduling. (3) Finally, the *Experiment Knowledge Module* enhances interpretability with well-structured documentation, enabling collaboration in large-scale experiments.

Though inspired by scientific research across disciplines, `Curie` focuses on experimentation in computer science that come with LLM-friendly interfaces (Anthropic, 2024; Yang et al., 2024). To evaluate `Curie`, we introduce an **Experimentation Benchmark** comprising 46 tasks of varying complexity across multiple computer science domains (§4). We derive these questions directly from influential research papers and practical open-source projects. Fig. 2 shows that `Curie` could reproduce, extend, and challenge existing research via rigorous experimentation. We benchmarked `Curie` (§5) against state-of-the-art agents like OpenHands (Wang et al., 2024d) and Magentic-one (Fourney et al., 2024). `Curie` achieves a 3.4× improvement in correctly answering experimental questions, underscoring `Curie`'s ability to automate experiments rigorously.

## 2 BACKGROUND

### 2.1 SCIENCE EXPERIMENTATION

Scientific experimentation often starts with researchers posing testable hypotheses based on their past results, domain knowledge, and intuition. This process then unfolds across three key stages: (1) *Experimental Design*, where researchers plan the controlled experiment by identifying variables, selecting methodologies, and outlining procedures to enhance reproducibility and validity; (2) *Experiment Execution*, where researchers set up the complex experiment environments and iteratively explore vast search spaces; and (3) *Data Documentation and Analysis*, where researchers systematically gather data, apply analytical techniques, and extract insights to validate or refine their hypotheses. This

process is iterative, as insights gained from data analysis often lead to the refinement of hypotheses, leading to subsequent rounds of these three steps.

## 2.2 RIGOR IN EXPERIMENTATION

Rigor is essential in scientific research, ensuring systematic, precise, and reliable findings (Armour et al., 2009). *If science isn't rigorous, it's reckless.* (Hofseth, 2018). Experimental rigor is grounded in three core principles (Gill & Gill, 2020):

**Methodical Procedure**: Experimentation must adhere to a principled and systematic methodology throughout all aforementioned stages, from hypothesis formulation to data documentation. Such a structured procedure ensures that no critical procedures are overlooked or performed incompletely, thereby preserving the integrity of the research.

**Reliability**: Every stage in the experimental pipeline—such as experiment design and environment setup—needs to be reliable and reproducible so that any final findings rest on solid ground. For instance, it encompasses correct variable identification, controlled experimental design, and rigorous code verification. By meticulously verifying each stage, reliability minimizes the risk of cascading errors, thereby ensuring that the results are trustworthy.

**Interpretability**: All processes and outcomes need to be clearly documented in a consistent manner. This makes it easier for researchers or agents to replicate experiments and understand results.

## 2.3 RELATED WORK

**AI Agents for Science.**   Prior work leveraged AI to accelerate scientific discovery (Berens et al., 2023; Kitano, 2021), focusing on various stages of the research lifecycle, including literature reviews (Agarwal et al., 2024; Tyser et al., 2024), brainstorming ideas (Gu & Krenn, 2024; Bran et al., 2024), hypothesis generation (Sourati & Evans, 2023; Zhou et al., 2024; Wang et al., 2024a; Qi et al., 2024) and data analysis (Hong et al., 2024a; Chen et al., 2024). However, experimentation—a critical, rigor-intensive step—remains underexplored. Existing agents for end-to-end scientific research (Schmidgall et al., 2025; Lu et al., 2024a; Yuan et al., 2025; Ghafarollahi & Buehler, 2024) rely on ad-hoc prompts to guide predefined workflows, from idea generation to paper writing. Their open-sourced frameworks often require experimental code to follow constrained, framework-specific formats, adding overhead and hindering their usability. These solutions mimic experimentation processes using multi-agent systems but lack systematic enforcement of a *methodical procedure*, *reliability*, and *interpretability*. Without these core principles, such agents struggle to deliver meaningful and reproducible results, limiting their practical utility in real-world scientific research. App. D.2 discusses their relation with `Curie`.

**AI Agent Task Benchmarks.**   A wide range of benchmarks have been developed to assess the capabilities of AI agents across diverse domains. Existing benchmarks primarily focus on logical reasoning (Cobbe et al., 2021; Hendrycks et al., 2021a; Bang et al., 2023), problem-solving (Hendrycks et al., 2021b; Frieder et al., 2023; Wang et al., 2024b; Sun et al., 2024a; Chevalier et al., 2024), knowledge retrieval tasks (Sun et al., 2024b) and machine learning training (Huang et al., 2024; Zhang et al., 2023; 2024a). These benchmarks evaluate agents on well-defined tasks that typically have clear, deterministic solutions (see App. D.1). In contrast, our benchmark focuses on experimentation, which requires a more rigorous and systematic approach beyond problem-solving. Experimental tasks require iterative hypothesis refinement, complex experiment setup and execution, and robust result interpretation. Our benchmark captures these challenges by evaluating AI systems on real-world experiments derived from influential research papers and widely adopted open-source projects.

## 3   CURIE: RIGOROUS EXPERIMENTATION

### 3.1 ARCHITECTURAL OVERVIEW

As shown in Fig. 3, `Curie` is composed of two types of LLM-based agents (an **Architect** Agent and a host of **Technician** Agents), sandwiched between them is our main innovation, the **Experimental Rigor Engine** that injects rigor throughout the experimental process.

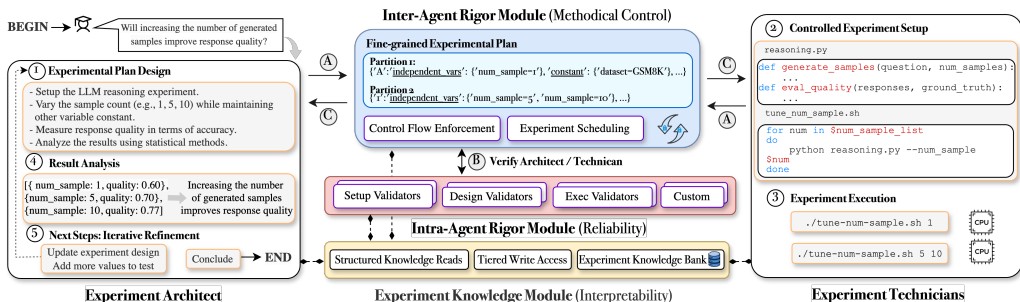

Figure 3: `Curie` workflow with an example task in LLM reasoning. The Architect designs high-level plans and reflects on findings. The Technician implements and executes the experiments based on the plans. Whenever an agent completes its action (step ①, ②, ③, ④, ⑤), the Experimental Rigor Engine (steps Ⓐ→Ⓑ→Ⓒ) validates the action, determines next steps, assigns tasks and maintains interpretable experimental progress, ensuring rigor throughout the entire process.

**High-level workflow.** Given an experimental question, our Architect will ① designs high-level *experimental plans* (e.g., defining hypotheses, variables), completing its turn. Our Inter-Agent Rigor Module (***Inter-ARM***) will Ⓐ intercept and enforce *methodical procedure*. Since the plan is new, it is broken into smaller partitions for finer-grained execution. *Inter-ARM* applies control flow policies to determine the next step for each partition. In this case, it decides to go through the Ⓑ the Intra-Agent Rigor Module (***Intra-ARM***) validation, which enhances *reliability* by verifying partition integrity (e.g., assessing relevance to the experimental question). Similarly, *Inter-ARM* repeats this process based on the validation results, eventually Ⓒ forwarding the partition to a Technician to ② set up the controlled experiment. The remaining steps are omitted for brevity, but at a high level, every agent action follows the same structured workflow: Ⓐ interception by *Inter-ARM*, Ⓑ validation by *Intra-ARM*, and Ⓒ forwarding to the next appropriate agent. Finally, all of the above components will make use of our **Experiment Knowledge Module** for storing and tracking experimental progress, providing *interpretability*. For example, the Architect stores refined experimental plans in a structured, metadata-enriched format, making them easier to analyze, track, and validate over time.

## 3.2  Intra-Agent Rigor Module - Reliability

Large-scale and long-running experiments involve complex, interdependent steps where early-stage errors can propagate and compromise final results. This is especially critical to LLM-based experimentation since: (1) LLM-based agents are prone to hallucination, and (2) experimental processes are inherently exploratory, requiring iterative refinements to hypotheses, setups, and designs in response to new or unexpected findings. Despite this, existing works (Lu et al., 2024a; Schmidgall et al., 2025) largely overlook the need for continuous validation throughout the experimental process. A naive approach is to perform end-to-end validation only after an experiment concludes. However, this lacks the ability to backtrack to intermediate stages, preventing error isolation and correction, and forcing researchers to either discard progress or rerun the entire experiment—an inefficient and costly approach. To address this, we introduce *Intra-ARM*, a validation module that verifies the assigned tasks of our Architect and Technicians step by step, improving reliability and reproducibility to align with the overarching experimental objectives. Inspired by process supervision (Lightman et al., 2023), *Intra-ARM* utilizes **modular validation**, where a suite of validators continuously verifies each stage of the experiment (Fig.3), so that errors can be proactively detected and addressed early. Moreover, *Intra-ARM*'s validators are extensible, allowing new ones to be incorporated as needed. We focus on two key validators here for brevity:

**Experimental Setup Validator.** This component (see App. H, Fig. 10) verifies that the experimental setup by our technicians aligns with the plan before execution, ensuring methodological soundness and logical consistency. Each enforced policy checks alignment within a specific part of the experiment setup. This includes (see App. H, Fig. 11a): (1) confirming the setup aligns with the experimental plan, including the research question and all specified variables (independent, dependent, and constant). (2) Analyzing all procedures for correct handling of input/output arguments; and detecting placeholders,

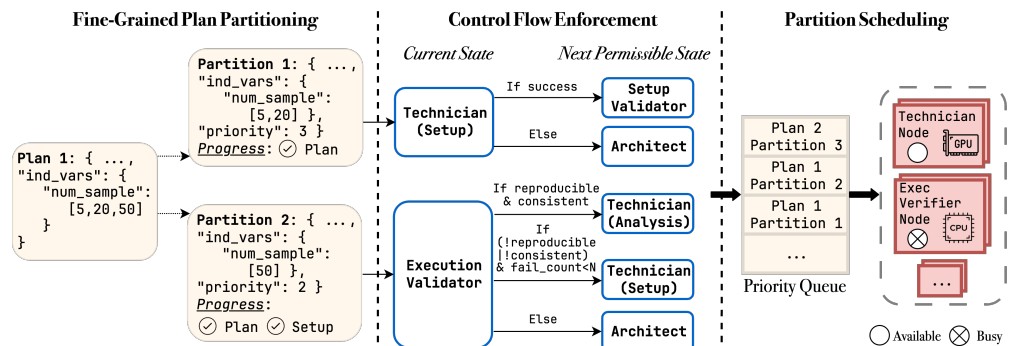

Figure 4: Simplified *Inter*-ARM workflow with a partition state snapshot.

hardcoded values, or incomplete variables to ensure meaningful results. (3) Checking that the setup documents all intermediate steps and results, including any identified issues for future analysis.

**Execution Validator.** Once the setup passes the experimental setup validator, this validator enhances reproducibility by executing it in a controlled and clean environment to detect and resolve potential errors, a sample of which is illustrated in App. H, Fig. 11. (1) *Error-Free Execution:* The setup is executed in a clean environment, verifying that it operates without errors. Any encountered errors are logged in detail, providing actionable feedback for debugging and iterative refinement. (2) *Reproducibility Checks:* The workflow is also run multiple times to enhance consistency in outputs and detect anomalies or hidden dependencies. Finally, the results are validated to ensure alignment with the experimental plan and compliance with predefined quality standards.

### 3.3 INTER-AGENT RIGOR MODULE - METHODICAL CONTROL

Experimental processes must follow a methodical procedure (§2.2) while balancing resource constraints (e.g., GPU availability), and experiment priorities. Traditional agentic conversational patterns (AutoGen, 2024)—such as naive LLM-based coordination, sequential, or round-robin execution—are thus ill-suited for such a workflow. To *ensure task coordination* and *optimize resource efficiency*, *Inter-ARM* enables seamless collaboration between our Architect, Technicians and *Intra*-ARM through three key functions presented in Fig. 4. We discuss each in turn.

**Fine-grained Plan Partitioning.** *Inter-ARM* first breaks down new complex experimental plans generated by the Architect into smaller, independent partitions: defined as a distinct subset of independent variable values within the plan. By creating smaller, self-contained tasks, this facilitates modular execution and enables parallelization, making experimentation more scalable. In addition, this enables our Architect to track intermediate progress and results, making real-time decisions as new insights emerge (e.g., reprioritizing partitions by updating their execution priority).

**Control Flow Enforcement.** This component ensures that transitions between our Architect, Technicians, and *Intra*-ARM follow a logical sequence aligned with the experimentation lifecycle. This is critical to maintaining consistent, error-free progress. Without structured coordination, tasks may be executed out of order or without necessary dependencies, leading to wasted effort and erroneous conclusions. For instance, it prevents Technicians from directly executing experiment setups before validation by *Intra*-ARM's setup validator, to reduce the risk of erroneous data propagation. This is done in two steps: (1) *State Evaluation*, which evaluates whether the current state of each partition (within an experimental plan) has been modified by an agent, e.g., a Technician who produced experimental results and recorded its progress via the Experiment Knowledge Module. (2) *Permissible State Transitions*, which produces a set of allowed state transitions for a partition based on its current state, e.g., newly produced experimental results for a given partition need to be validated by *Intra-ARM* first. It also gathers relevant context that would be useful if the transition were to be executed. This state transition information will be consumed by our scheduler (defined below).

**Partition Scheduling.** Large-scale experiments can be resource-intensive and time-consuming, requiring careful scheduling and prioritization of tasks to improve efficiency. Our scheduler currently utilizes three knobs for partition scheduling: (1) partition execution priorities set by our Architect, (2) allowed partition state transitions, and (3) the availability of our agents (that may be busy handling other partitions). Overall, this adaptive scheduling strategy enables large-scale experimentation by improving resource efficiency while adhering to methodical experimental procedures.

### 3.4 EXPERIMENT KNOWLEDGE MODULE - INTERPRETABILITY

Interpretability is fundamental to experimentation—not only for scientific accountability but also for effective experiment management. Specifically, all other components within `Curie` require this for real-time visibility, enabling informed decision-making, efficient troubleshooting, and adaptability as new insights emerge. A naive approach would be to delegate experimental knowledge management entirely to LLM-based agents. However, LLMs alone are ill-suited for this task for two reasons: (1) *Inconsistent Reads*: LLMs have inconsistent recall and are prone to forgetting (Xu et al., 2024). Without a structured and verifiable record of experimental progress, they may retrieve outdated, irrelevant, or hallucinated information, leading to misinterpretations, flawed conclusions, and compounding errors over time. (2) *Inconsistent Writes*: LLMs tend to hallucinate, particularly when managing large-scale experimental data. This lack of structured control risks corrupting experimental records, propagating inaccuracies, and ultimately compromising the integrity of the experimentation process. Unlike databases, LLMs do not inherently track provenance (Hoque et al., 2024), making it difficult to reconstruct how conclusions were reached. We address these two challenges in turn:

**Structured Knowledge Reads.** This mechanism organizes experimental progress in a structured format. The process begins by restructuring new experimental plans that were written by our Architect into an enriched format with critical metadata—such as setups, execution status, and results. Subsequent modifications to any part of the plan are recorded as a time machine (see App. H, Fig. 12) for experimental progression, maintaining a structured, DAG-like history of changes. This historical record captures hypotheses tested, variable changes, and the reasoning behind key decisions. By preserving this evolution, `Curie` can reconstruct past states, trace decision rationales, and diagnose issues with greater precision.

**Tiered Write Access.** To maintain experimental integrity and minimize the risk of errors, the interface enforces a tiered write access policy that restricts and validates updates made to the experimental plan. This ensures that each component can only modify the portions of the plan they are responsible for, while all changes undergo rigorous validation. For example, Technicians are permitted to append experimental results to their assigned partitions but cannot modify unrelated sections of the plan. Similarly, architects have broader write access, including the ability to create or remove entire partitions, but their modifications are still constrained to specific attributes, such as updating variable values or marking partitions for re-execution. Every write operation is validated before being committed to the knowledge bank. This process ensures proper structuring of inputs and enforces semantic integrity (e.g., that result file paths are valid). If errors are detected, the system returns concise error messages, enabling agents to quickly identify and resolve issues. Through this, `Curie` enhances robustness and error resistance in collaboration.

## 4 EXPERIMENTATION BENCHMARK

We design a novel benchmark to stress test `Curie`'s ability to automate experiments while enforcing rigor in front of real-world challenges. As shown in App. E Table 4 (with full details in App. F), our benchmark consists of 46 tasks across 4 domains within computer science (reasoning in App. D.3). Our tasks are derived directly from **real-world influential research papers** and use-cases within **popular open-source projects**. We have open-sourced our benchmark alongside the agent framework.

### 4.1 EXPERIMENT-CENTRIC TASK DESIGN

Instead of treating tasks as isolated problems with fixed solutions, we structure each task as a full experimental process. This means that tasks require hypothesis formation, iterative refinement, and rigorous validation, mirroring real-world experiment workflows rather than one-shot problem-solving.

Table 1: Main benchmark results in terms of four metrics introduced in §5. We aggregate and average the success rate among all tasks within each domain. The final row presents the weighted average, computed based on the number of tasks in each domain. The standard error of success rate across random trials are shown in App. C.2 Table 3

| Domain | Curie | | | | OpenHands | | | | Microsoft Magentic-One | | | |
|---|---|---|---|---|---|---|---|---|---|---|---|---|
| | Des. | Exe. | Alig. | Con. | Des. | Exe. | Alig. | Con. | Des. | Exe. | Alig. | Con. |
| LLM Reason. | 98.3 | 83.3 | 76.7 | 44.9 | 86.7 | 24.6 | 36.7 | 14.2 | 72.0 | 9.3 | 14.0 | 6.7 |
| Vector DB | 97.8 | 71.7 | 77.2 | 25.6 | 85.0 | 48.3 | 52.3 | 11.7 | 85.0 | 6.4 | 63.6 | 0.0 |
| Cloud Comp. | 100.0 | 92.7 | 96.9 | 32.3 | 96.9 | 25.2 | 49.2 | 5.0 | 95.0 | 6.3 | 33.8 | 0.0 |
| ML Training | 95.2 | 66.7 | 39.3 | 41.7 | 63.1 | 24.3 | 16.7 | 5.7 | 90.0 | 2.9 | 25.7 | 0.0 |
| Weighted **Avg.** | 97.9 | 78.1 | 73.4 | 36.1 | 83.6 | 32.4 | 40.2 | 10.5 | 82.9 | 6.8 | 35.2 | 2.3 |

The process begins with distilling high-level contributions from research papers (e.g., theoretical insights or empirical findings), or core system behaviors from open-source projects (e.g., the interplay between configuration parameters and performance). These insights are then translated into testable questions framed with explicit configurations, metrics, and expected outcomes. Ground truth data is derived from published results or official benchmarks provided by open-source projects. We use these findings to design tasks with three key components:

**1. Experiment Formulation:** Each task specifies the (a) Experiment Question (e.g., optimizing performance); (b) Practical constraints (e.g., resource budgets); (c) High-level Setup Requirements - Contextual details such as datasets, and experimental environments. This framing ensures that tasks are open-ended, requiring iterative exploration rather than one-shot solutions.

**2. Experimental Context:** To ensure agents correctly interpret and execute tasks, the benchmark provides detailed context for each question. This includes: (a) Domain Knowledge – Background information essential for interpreting the problem. (b) Starter Code & Tools – Predefined scaffolding to simulate real-world research workflows.

**3. Ground Truth:** This is defined in two key areas: (a) *Experimental Design*: Does the agent correctly formulate the experiment, identifying relevant variables and methodologies? (b) *Result Analysis:* Does the agent correctly interpret findings, and justify its conclusions? We outline the expected outcomes or acceptable solution ranges.

## 4.2 EXPERIMENTAL COMPLEXITY

Experimental research varies in complexity across different dimensions. Our benchmark reflects this by structuring tasks into a hierarchical framework, assessing an agent's ability to handle increasingly sophisticated experimentation tasks. Unlike standard benchmarks that classify tasks by a single difficulty metric (e.g., easy, medium, hard), ours structures complexity along experiment-driven dimensions (detailed definitions in App. A):

*1). Design Complexity:* The complexity of structuring an experiment (e.g., requiring hypothesis refinement), including defining the scope of exploration, selecting key variables, and structuring parameter spaces—ranging from discrete to continuous and from sparse to dense configurations.

*2). Experiment Setup Complexity:* The difficulty of initializing and configuring the experimental environment, from simple predefined setups to intricate dependencies requiring multi-step configuration.

*3). Relationship Complexity:* The interactions between variables and outcomes, from simple linear dependencies to complex non-monotonic relationships.

*4). Experiment Goal Complexity:* The number of competing objectives and trade-offs involved, from single-metric optimization to multi-objective balancing under constraints.

## 5 EVALUATION

We evaluate `Curie` using our experimentation benchmark, which consists of 46 research tasks spanning varying complexity levels across four key domains (§4). To enhance statistical robustness,

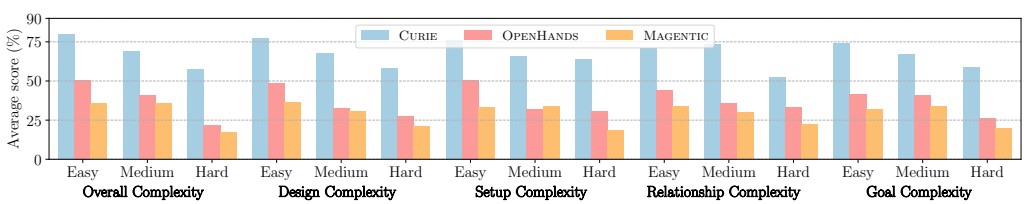

Figure 5: Average scores across different complexity dimensions at varying difficulty levels. `Curie` outperforms baselines consistently, with performance generally dropping as complexity increases.

each task is executed independently for five trials for each of our baselines (below) and `Curie`, and we report the average performance across these trials. Apart from our main results described in §5.1, our evaluation includes our case studies (Fig. 2 and App. B), and additional results (App. C.1).

**Baselines.** We compare `Curie` with two state-of-the-art AI agents as our *baselines*: Open-Hands (Wang et al., 2024d), a top-performing coding agent, and Microsoft Magentic (Fourney et al., 2024), a generalist multi-agent system. These baselines were selected because our benchmark primarily focuses on coding-related tasks within computer science, where both models demonstrate strong performance, with the expectation that Magentic, as a generalist multi-agent system, may be able to generalize to experimental tasks too. To ensure fairness, each baseline is provided with a detailed system prompt instructing them to act as a professional experimenter (see App. G.1). All baselines and `Curie` utilize GPT-4o as the underlying LLM.

**Performance Metrics.** We assess performance using four key metrics, each evaluated as a binary score per task, ensuring rigor at every stage of the experimentation process:

*1. Experiment Design* – Ability to design the high-level experiment plan to address the question.

*2. Execution Setup* – Ensuring that the generated code (experiment setup) is executable and produces consistent results across multiple runs.

*3. Implementation Alignment* – Faithfulness of the experimental setup with the proposed plan.

*4. Conclusion Correctness* – Accuracy in reflecting the ground truth answer to the question.

**Evaluator.** We employ an LLM judge (Zheng et al., 2023) for straightforward verification such as checking design, setup and conclusion, where the ground truth is known. However, we manually assess the implementation alignment, as detecting semantic discrepancies between the intended methodology and code is non-trivial (reasoning in App. D.5). To ensure accuracy, we also verify the LLM judge's assessments by cross-checking a subset of its evaluations against expert annotations, measuring agreement rates, and refining the judge system prompt. Details of the evaluation prompts are provided in App. G.2. This hybrid evaluation approach enables reliable and scalable assessment of experimentation performance.

## 5.1 BENCHMARK PERFORMANCE

Table 1 shows aggregated success rates across all performance metrics and benchmark task domains.

**Performance Breakdown By Metric.** Across all four metrics, `Curie` consistently outperforms the baselines, demonstrating the benefits of our Experimental Rigor Engine in improving experimentation performance. (i) For experiment design correctness, all frameworks perform well since the current tasks are relatively straightforward and do not require iterative refinement. However, for more complex research tasks, `Curie` holds an advantage by dynamically refining hypotheses based on intermediate observations, whereas baselines rely on static planning. Our experimental knowledge module further enhances performance by improving recall and adaptation. (ii) For execution setup and implementation alignment, `Curie` demonstrates higher reliability, as *Intra-ARM* proactively validates and corrects execution steps, while *Inter-ARM* guarantees that we follow methodical task transitions. This results in particularly strong execution setup performance, from 66.7% to 92.7%.

OpenHands (with 32.4% and 40.2%), as a coding-specialized agent, outperforms Magentic in this aspect. However, it still struggles with incomplete or erroneous setups, including getting stuck in loops, syntax errors, logic mistakes, and unresolved dependencies—leading to execution failures in complex environments. Magentic, in particular, performs poorly in locating the correct files in the task starter file and handling script input/output. (iii) Finally, for conclusion correctness, its accuracy is largely constrained by earlier errors, as conclusions rely on the correctness of experimental results. However, `Curie` maintains a strong lead due to its Experiment Knowledge Module, which systematically documents experimental results for structured data analysis. This enables `Curie` to achieve a significantly higher conclusion score of 36.1%, compared to 10.5% for OpenHands and 2.3% for Magentic. While Magentic demonstrates relatively decent alignment, it struggles to translate this into meaningful conclusions because of previous cascading errors.

**Performance Breakdown By Domain.** Across all four task domains, `Curie` consistently outperforms the baselines, demonstrating `Curie`'s ability to adapt to different research domains. (i) First, for LLM reasoning tasks, `Curie` performed exceptionally well, achieving the highest conclusion accuracy at 44.9%. OpenHands had its best performance in this category (14.2%), while Magentic attained its only non-zero score of 6.7%. We attribute this to the inherent intuitiveness of conclusions for our tasks in this domain. (ii) For Vector DB tasks, both OpenHands and Magentic achieved their highest alignment scores—52.3% and 63.6%, respectively—likely due to the familiarity of the task. Alignment was also easier given the availability of well-established open-source benchmarks and shorter execution runs, which provided faster feedback. (iii) For Cloud Computing tasks, `Curie` outperformed OpenHands significantly in all aspects (e.g., $6.5\times$ the conclusion accuracy). This is because these tasks often involve long-running experiments, which requires robust execution tracking and dynamical experimentation workflows adjustment based on partial results. (iv) Finally, for ML Training tasks, all agents underperformed in alignment and execution as the detailed environment setup instructions are not provided for these tasks. Despite this, `Curie` can figure out the correct setup by reflection and refinement, achieving a $7.3\times$ higher conclusion accuracy than OpenHands.

**Performance Breakdown by Complexity.** Next, we analyze how each framework performs as we increase difficulty within each complexity dimension. Fig. 5 reports the aggregated performance score, computed as the average across all four evaluation metrics. We observe that increasing complexity difficulties across all dimensions correlates with a decline in performance across all agents. However, the rate of degradation varies across complexity types and agent architectures. Notably, Magentic consistently underperforms across all complexity levels, highlighting the robustness of our complexity-based difficulty scaling in distinguishing agent capabilities. Further, we observe a sublinear decline in performance as task complexity increases, suggesting that our hardest tasks could be made even more challenging. Despite this, our current results demonstrate `Curie`'s capabilities, supported by our case studies. Exploring the limit of experimentation difficulty and its impact on model performance remains an open direction for future work.

In summary, our findings underscore the importance of rigorous evaluation across experimentation stages, shedding light on each framework's strengths and limitations under varying conditions.

## 6 CONCLUSION AND FUTURE WORK

We introduced `Curie`, an AI agent designed to automate and enhance the rigor of scientific experimentation. Central to its design is the Experimental Rigor Engine, which enforces methodical control, reliability, and interpretability. To assess `Curie`'s effectiveness, we developed a new Experimentation Benchmark featuring real-world research challenges. Our empirical evaluation, comparing `Curie` against state-of-the-art agents, demonstrated its capability to automate rigorous experimentation.

We hope `Curie` inspires further advancements toward fully autonomous and rigorous experimentation in the era of AI agent-driven scientific research. Several open research challenges remain: For instance, adapting `Curie` for interdisciplinary research requires accommodating domain-specific methodologies, uncertainty control, and extended time scales, such as long-term biological studies (Hilty et al., 2021). Moreover, enabling knowledge reuse (Wang et al., 2024e) across experiments could enhance efficiency and further accelerate discovery.

## 7 REPRODUCIBILITY STATEMENT

All code and data supporting our work are available through an anonymous repository at `https://anonymous.4open.science/r/Curie-689B/`, and will be open-sourced upon acceptance. Details of our benchmark are provided in App. F and §4, while details of our system architecture are described in §3. The benchmark complexity and task selection process are described in App. A and App. D.3, respectively. `Curie`'s system prompts and evaluation prompts are provided in App. G.1 and App. G.2, respectively. Various case studies for `Curie` are described in App. B. Finally, additional evaluation results and analyses are provided in App. C.

## 8 THE USE OF LARGE LANGUAGE MODELS (LLMS)

In accordance with the ICLR 2026 guidelines on LLM usage, we disclose that LLMs were used solely for grammar and style checking during the preparation of this manuscript. No LLMs contributed to research ideation, experimental design, analysis, or substantive writing.

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

Table 2: Descriptions of various complexity levels for experiments across multiple dimensions.

| Complexity Dimension | Level | Description and Example |
|---|---|---|
| Experiment Setup | Easy | Straightforward setup with minimal dependencies. Example: Running an inference script on local hardware. |
| | Med. | Moderate setup involving multiple components. Example: Setting up a VM cluster and distributing workloads. |
| | Hard | Complex setup requiring multiple dependencies and external configurations. Example: Setting up a distributed system with networking, storage, and inter-region communication. |
| Design | Easy | Well-defined experiments with few variables, and simple parameter spaces. |
| | Med. | Requires a moderate number of multiple key variables; with a mix of discrete and continuous parameters. |
| | Hard | Involves complex variable interactions, and densely structured parameter spaces requiring adaptive exploration. |
| Experiment Goal | Easy | Single metric with a clear, measurable goal and no significant trade-offs. Example: Success rate for a configuration. |
| | Med. | Multiple objectives, with moderate trade-offs but relatively independent goals. Example: Balancing cost and latency. |
| | Hard | Conflicting objectives with high interdependencies, requiring sophisticated optimization and rigorous validation. Example: Minimizing cost while ensuring latency under 100ms and CPU utilization above 80%. |
| Relationship | Easy | Linear relationships. Example: Performance scales linearly with the number of CPUs. |
| | Med. | Nonlinear but monotonic relationships: e.g., sublinear, logarithmic. Example: Diminishing returns in performance as more CPUs are added. |
| | Hard | Non-monotonic or stochastic dependencies. Example: Performance fluctuates due to unpredictable network interference. |
| Overall | Easy | If none of the below hold. |
| | Med. | At least 2 dimensions are medium, or if only 1 dimension is hard with 1 other dimension being medium. |
| | Hard | At least 2 dimensions are hard. |

## A  CURIE BENCHMARK COMPLEXITY EXPLANATION

We describe in detail our complexity level definitions in Table. 2.

## B  CASE STUDIES FOR CURIE

We provide two example case studies for LLM reasoning tasks that `Curie` was able to extend from the paper *The Impact of Reasoning Step Length on Large Language Models* (Jin et al., 2024).

In Fig. 6a, the objective of this experiment is to examine whether different models exhibit varying accuracy levels based on the number of reasoning steps. The experiment maintains constant variables, including the dataset (`last_letters`), the method (`auto_cot`), and the evaluation metric (`accuracy`). The independent variables include the model type (`gpt-4o-mini` vs. `gpt-4o`) and the number of reasoning steps (1, 2, 3, 4, 5, 6, 10), while the dependent variable is the model's accuracy. The experiment consists of a control group and experimental groups. The control group uses `gpt-4o-mini` with a single reasoning step to establish a baseline accuracy. The experimental groups involve testing `gpt-4o-mini` with reasoning steps ranging from 2 to 10 and `gpt-4o` with reasoning steps from 1 to 10. The results will help determine whether reasoning step variations impact accuracy differently across models.

`Curie` extends the original investigation by examining whether different LLMs exhibit varying accuracy using GPT-4o and GPT-4o-mini. While the original work primarily focused on general

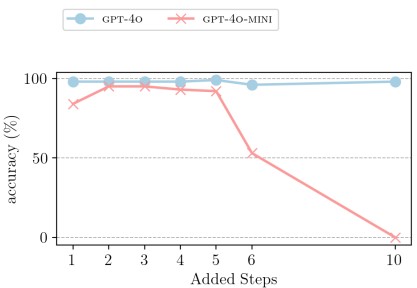 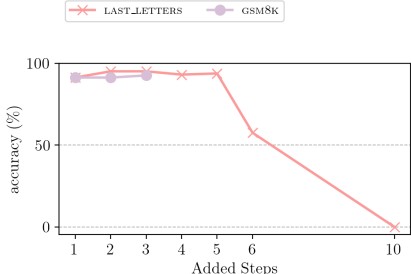

(a) Question 6: "Does the optimal number of reasoning steps vary across different LLMs?"

(b) Question 8: "What is the relationship between the complexity of a task (e.g., as measured by the number of logical inferences or mathematical operations needed) and the optimal length of the reasoning chain?"

Figure 6: Case studies on LLM reasoning tasks.

trends, `Curie` establishes a structured experimental framework that includes both control and experimental groups and introduces a new focus on optimal reasoning steps. This refinement provides a more nuanced understanding of how reasoning steps affects accuracy across different LLM architectures.

In Fig. 6b, the objective of this experiment is to examine the relationship between task complexity and the optimal length of reasoning chains in large language models (LLMs). The experiment maintains constant variables, including the model (`gpt-4o-mini`), the method (`auto_cot`), and the environment setup (OpenAI credentials and a Conda environment). The independent variable is the number of reasoning steps, controlled through different demo files, while the dependent variable is the model's accuracy, as reported in the log files. The experiment consists of a control group and experimental groups. The control group uses the `gsm8k_1` demo file with a single reasoning step to establish a baseline accuracy. The experimental groups involve testing `gsm8k` with reasoning steps from `gsm8k_2` and `gsm8k_3`, and `last_letters` with reasoning steps ranging from `last_letters_1` to `last_letters_10`. The results will help determine whether task complexity influences the optimal number of reasoning steps required for maximizing accuracy in LLMs.

`Curie` extends the scope by analyzing how task complexity relates to the optimal length of reasoning chains. This study differentiates between problem types (e.g., logical inference and mathematical operations) and systematically evaluates the effect of reasoning step count within different datasets (`gsm8k` and `last_letters`). By introducing controlled experimental conditions, `Curie` enables a more detailed exploration of how task complexity interacts with reasoning steps to optimize model performance.

## C    EXTENDED EVALUATION

### C.1    FINE-GRAINED PERFORMANCE BREAKDOWN BY INDIVIDUAL METRICS

We detail fine-grained breakdowns for each of our performance metrics mentioned in §5. Here we observe the general trend that increasing complexity across all dimensions causes reductions in average metric scores, as shown in Fig. 7, Fig. 8 and Fig. 9, respectively. In particular, we observe that conclusion scores are most heavily affected as complexity increases across dimensions, reaching 0% on many occasions for Magentic in particular. For design complexity on the other hand, we observe that we're able to maintain a relatively high average score across all baselines and `Curie`, but this tapers down as the difficulty increases across dimensions.

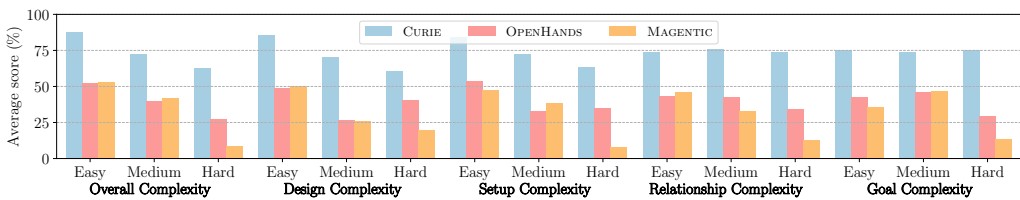

Figure 7: Average alignment scores across different complexity dimensions at varying difficulty levels for `Curie`, OpenHands, and Magentic. `Curie` outperforms the others consistently, with performance generally dropping as complexity increases.

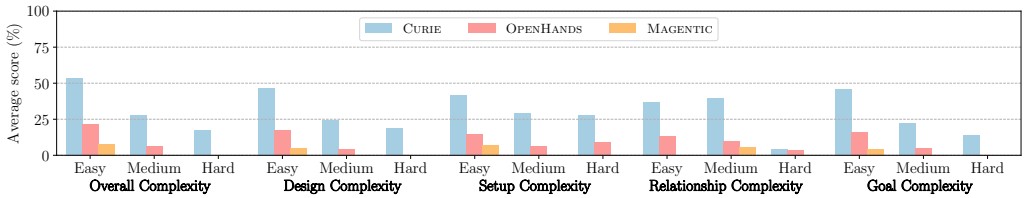

Figure 8: Average conclusion scores across different complexity dimensions at varying difficulty levels for `Curie`, OpenHands, and Magentic. `Curie` outperforms the others consistently, with performance generally dropping as complexity increases.

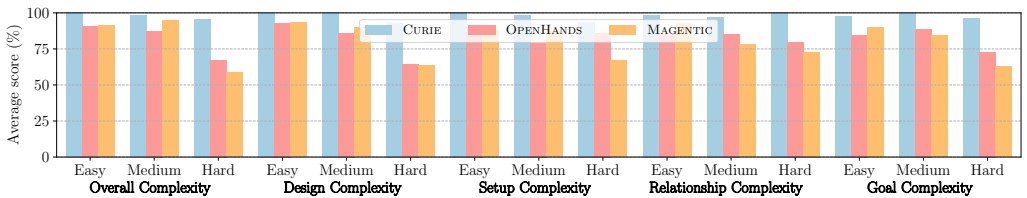

Figure 9: Average design scores across different complexity dimensions at varying difficulty levels for `Curie`, OpenHands, and Magentic. `Curie` outperforms the others consistently, with performance generally dropping as complexity increases.

Table 3: Standard error across random trials in terms of four metrics introduced in §5.

| Domain | Curie | | | | OpenHands | | | | Microsoft Magentic-One | | | |
|---|---|---|---|---|---|---|---|---|---|---|---|---|
| | Des. | Exe. | Alig. | Con. | Des. | Exe. | Alig. | Con. | Des. | Exe. | Alig. | Con. |
| LLM Reason. | 6.3 | 9.8 | 8.6 | 8.6 | 12.5 | 6.7 | 7.5 | 5.0 | 8.2 | 4.2 | 5.1 | 6.1 |
| Vector DB | 2.2 | 7.3 | 6.8 | 8.2 | 7.8 | 8.5 | 9.0 | 5.7 | 7.8 | 3.5 | 8.5 | 0.0 |
| Cloud Comp. | 0.0 | 4.5 | 2.9 | 13.0 | 2.9 | 4.8 | 8.9 | 4.7 | 3.1 | 5.9 | 14.7 | 0.0 |
| ML Training | 4.4 | 8.9 | 14.6 | 16.2 | 10.9 | 11.4 | 7.5 | 3.4 | 6.7 | 2.6 | 8.5 | 0.0 |
| Weighted **Avg.** | 97.9 | 78.1 | 73.4 | 36.1 | 83.6 | 32.4 | 40.2 | 10.5 | 82.9 | 6.8 | 35.2 | 2.3 |

## C.2 STANDARD ERROR ACROSS RANDOM TRIALS

To demonstrate the statistical significance of the results presented in Table 1, we present the standard error of the results across random trials. We evaluated `Curie` and our 2 baselines across 46 tasks with 5 independent trials each, yielding a total of 230 data points per framework. The number of trials conducted is consistent with related benchmarks; for instance, MLAgentBench ran 8 trials per task, while ScienceAgentBench ran 3 trials per task. Here, we compute standard errors of the mean pass rate across tasks, treating each task's average score over its 5 trials as one data point.

# D DISCUSSION

## D.1 RELATED BENCHMARKS

Our benchmark is necessary as there is currently no benchmark that captures the true nature of experimentation as practiced in real-world scientific settings. This gap exists because experimental tasks go beyond analyzing static datasets or single-step solutions—they require thoughtful design evaluation, complex setup procedures, and iterative reasoning and empirical testing to arrive at valid conclusions. Prior scientific benchmarks differ from ours: for instance, SciBench (Wang et al., 2024c) emphasizes scientific reasoning, such as mathematical problem-solving, which is categorically different from experimental inquiry. SciCode (Tian et al., 2024) targets domain-specific code generation for simple functions. BLADE (Gu et al., 2024) performs statistical analysis on fixed datasets or environments. In contrast, our benchmark includes tasks that require models to autonomously curate data. BixBench (Mitchener et al., 2025), a contemporaneous bioinformatics benchmark, explores open-ended tasks lacking clear optimization metrics, and we look forward to integrating it into our framework. Existing ML training benchmarks such as those mentioned in Agent K (Grosnit et al., 2024) typically provide preconfigured environments, skipping essential but potentially complex experiment setup procedures (e.g., installation of packages, dependency management) that must be completed first. In contrast, our benchmark mirror realistic experimentation scenarios, where researchers are required to build and configure their experimental environments from scratch.

## D.2 RELATED AGENTS

Our view is that the existing iteration of Deep Research (DR) is complementary to `Curie`, and most ideally suited for the hypothesis-generation phase prior to experimentation. According to its official description, DR is designed to "find, analyze, and synthesize hundreds of online sources", optimized for "web browsing and data analysis", and leverages "browser and Python tool use" to "expedite complex, time-intensive web research". As an example, our cloud computing experiments would benefit from using DR to efficiently gather detailed information from the web about specific machine configurations and associated costs, followed by `Curie` for the subsequent experimentation phase, which involves building, configuring, and interacting directly with remote cloud machines. We also envision `Curie` to be used as an experiment module within AI Scientist (Lu et al., 2024b), as it's current experimentation module is composed of simple LLM prompts. Also, Aviary (Narayanan et al., 2024) serves primarily as a gymnasium focused on providing abstractions and interfaces (e.g., building scenario-specific environments) for scientific agent development through learning. We can leverage Aviary's learning capabilities within specialized tasks and then apply `Curie` to enforce rigor.

Our baseline agents are representative for our domains. OpenHands is one of the strongest coding agents available, that has seen integration with various scientific and ML benchmarks, including BioCoder (Tang et al., 2024b), DiscoveryBench (Majumder et al., 2024), and ML-Bench (Tang et al., 2024a). Moreover, we include a strong generalist multi-agent system (Magentic-One) by Microsoft, which has seen strong performance on e.g., GAIA (Mialon et al., 2023). Finally, we ensure fairness in all comparisons by standardizing the evaluation setup: all agents are tested on the same tasks under identical conditions, and use the same underlying model configuration.

## D.3 BENCHMARK TASK SELECTION

Our benchmark comprises 46 scientific tasks selected to reflect the diversity and complexity of real-world experimentation. These include experiments directly extracted from research papers, capturing well-defined hypotheses, configurations, and evaluation criteria. We also include ML training tasks adapted from benchmarks such as MLAgentBench, which cover canonical problems like image classification, sentiment analysis, and Kaggle competitions. To reflect modern scientific workflows, we incorporate cloud computing tasks that require remote environment setup and interaction with external systems—scenarios commonly encountered in real experiments but rarely addressed in existing benchmarks. Additionally, we include vector indexing tasks (e.g., Faiss-based) that require agents to navigate trade-offs between recall, memory usage, and latency, analogous to real-world scientific challenges like tuning experimental conditions to balance yield, purity, and time.

Table 4: Experimentation benchmark overview. (E for Easy, M for Medium, H for Hard)

| Domain | Complexity | | | Description | Sources |
|---|---|---|---|---|---|
| | E. | M. | H. | | |
| LLM Reasoning | 4 | 5 | 7 | Investigates strategies for scaling test-time computation in LLMs. | Research papers: (Brown et al., 2024), (Jin et al., 2024). |
| Vector Indexing | 6 | 6 | 3 | Examines efficient vector indexing methods, analyzing its trade-offs. | Open-source repositories: Faiss (Douze et al., 2024) |
| Cloud Computing | 2 | 4 | 2 | Optimize various cloud setups. | Cloud providers: AWS |
| ML Training | 3 | 3 | 1 | Optimize ML training pipelines. | Benchmark: (Huang et al., 2024), (Hong et al., 2024b) |

Collectively, these tasks were chosen to evaluate an agent's ability to handle both conceptual rigor and operational complexity in automated experimentation.

## D.4 ABLATION STUDY

Performing ablation study by masking away `Curie` components is challenging in practice. To start with, the Rigor Engine is integral to `Curie` 's functionality, making isolated ablations challenging without fundamentally disrupting the experimentation process. Our logging analysis reveals that even at the initial step—formulating the experiment design plan—the Intra-Agent Rigor Module is critical, requiring multiple refinements to ensure a structured plan with essential elements like constant, dependent, and independent variables. Without this module, the design lacks the necessary format and rigor, rendering subsequent steps—like execution and analysis—unfeasible or misaligned.

In regards to our inter-agent rigor module and experiment knowledge module, they are necessary and fundamental components of rigor, as they are meant to guarantee methodical control and interpretability; in other words, it is not about the magnitude of their contribution to accuracy, but their ability to provide guarantees that matters. For instance, our knowledge module provides, among other things, a "time machine" view into the experiment—allowing users to trace exactly what occurred, when it happened, and how each fine-grained decision was made. This is crucial not only for interpretability but also for validating and reproducing experimental outcomes. Our inter-agent module, among other things, ensures that decisions are not made in isolation, e.g., each agent decision must be checked by an Intra-Agent rigor policy before proceeding, reducing the risk of spurious outcomes and enforcing a higher standard of internal consistency across the experimental pipeline.

## D.5 MANUAL EFFORTS

We manually assess the implementation alignment, as detecting semantic discrepancies between the intended methodology and code is non-trivial. We've noticed that the LLM judge can fail when the task requires a complex setup, or domain-specific understanding. As an example, the LLM judge may fail, for instance, in understanding that correctly implementing one of our cloud questions involves many intricate steps including instantiating a machine using specific AWS CLI commands, provisioning a unique key pair using openssl before attaching it, deploying traffic simulators on top of the machine, etc.

# E BENCHMARK COMPOSITION.

The composition of our benchmark is provided in Table. 4.

## F  BENCHMARK DETAILS.

| Domain | Question | Complexity | | | | |
|---|---|---|---|---|---|---|
| | | Design | Relat. | Goal | Setup | Overall |
| LLM Reasoning | How does the number of generated samples per question impact the overall success? | Easy | Easy | Easy | Easy | Easy |
| | What is the mathematical relationship between the number of generated samples per question and the overall success rate? For instance, does the rate of success scale linearly, quadratically, or follow another pattern as the number of generated samples increases? | Easy | Medium | Easy | Easy | Easy |
| | Considering that a larger, more capable model (e.g., gpt-4o) costs significantly more per query compared to a smaller model (e.g., gpt-4o-mini), would it be feasible to use the smaller model, sample more responses, and achieve comparable rate of success while being more cost-effective? | Medium | Medium | Medium | Easy | Medium |
| | To achieve 80% success rate for gsm8k task, what is the most cost-effective configuration? Specifically, which model (gpt-4o-mini or gpt-4o) should be used, and how many samples per question should be generated to minimize cost? You will need to test at least 4 samples sizes, and make sure to test each of the chosen samples sizes on both gpt-4o-mini and gpt-4o. | Hard | Medium | Hard | Hard | Hard |
| | How does varying the sampling temperature affect the diversity and quality of responses when using a fixed number of samples? | Hard | Hard | Hard | Medium | Hard |
| | One approach to scaling language model inference is to repeatedly sample candidate solutions from the model and aggregate them using majority voting. How does the number of samples impact the overall accuracy on the GSM8K task? | Medium | Hard | Easy | Medium | Medium |
| | How effective is paper's methodology to scale test-time compute, as repeated sampling in LLMs often leads to duplicate answers? | Medium | Medium | Easy | Medium | Medium |
| | Will increasing the number of reasoning steps in a Chain of Thought (CoT) prompt improve LLM accuracy up to a saturation point? | Hard | Hard | Medium | Medium | Hard |
| | Does the optimal number of reasoning steps for multi-step reasoning tasks vary based on the problem type (e.g., mathematical and logic problems)? | Medium | Medium | Hard | Hard | Hard |
| | Can the accuracy impact of different prompting methods like Zero-shot and Auto-CoT be systematically improved by varying the number of reasoning steps without adding new content in a tightly controlled experiment setting, by using methods such as adding sentences that restate the question to increase steps? | Easy | Medium | Easy | Easy | Easy |
| | How does the impact of an incorrect step on overall LLM performance vary across different task types, such as process-oriented tasks versus symbolic reasoning or logic tasks? | Hard | Medium | Hard | Medium | Hard |
| | What is the optimal number of reasoning steps for different types of tasks to maximize accuracy while minimizing computational cost? | Medium | Medium | Easy | Medium | Medium |
| | Does the optimal number of reasoning steps vary across different LLMs [GPT-4o, GPT_4o-mini], and if so, what is the nature of that relationship? | Hard | Medium | Easy | Medium | Medium |
| | How do different methods of expanding reasoning steps (e.g., repeating the question, self-verification, making equations) affect the model's accuracy, and are some expansion strategies more effective than others? | Hard | Medium | Easy | Hard | Hard |
| | What is the relationship between the complexity of a task (e.g., as measured by the number of logical inferences or mathematical operations needed) and the optimal length of the reasoning chain? | Easy | Medium | Easy | Easy | Easy |
| | How does the position of an incorrect step within the reasoning chain affect the overall outcome? Is an early error more detrimental than a later one? | Hard | Medium | Medium | Hard | Hard |
| | Considering that larger models generally perform better, would it be more cost-effective to use a smaller model with longer reasoning chains or a larger model with fewer steps for a given level of accuracy? | Hard | Medium | Medium | Hard | Hard |
| Vector Indexing | What is the relationship between query latency for the SIFT1M dataset and efSearch values with the HNSW index? Use a fixed value of k=10, M=32, efConstruction=40. | Easy | Easy | Easy | Easy | Easy |
| | What is the effect of varying M (number of neighbors per node) on the memory usage, recall, and query latency for the SIFT1M dataset with the HNSW index? Use varying M values of 16, 24, 32. Use fixed values of k=10, efConstruction=40. | Easy | Medium | Medium | Easy | Medium |
| | What is the optimal combination of M and efSearch to minimize memory usage while maintaining a recall of at least 90%? Use k=10, efConstruction=40, and use varying M values of 16, 24, 32. efSearch is not a parameter that you need to touch. | Easy | Easy | Medium | Easy | Easy |
| | What is the effect of parallelism (via omp_set_num_threads. You need to modify bench_hnsw.py to accept and use this parameter properly) on recall and latency for the SIFT1M dataset with a fixed efSearch=100, k=10, M=32, efConstruction=40 | Easy | Easy | Easy | Medium | Easy |
| | What is the highest recall that can be achieved on the SIFT1M dataset with an HNSW index while keeping query latency under 5ms? Report the optimal configuration. Use a fixed k value of 10, use varying M values of 16, 24, 32, use varying efConstruction values of 40, 50, 60. In total, there should be 9 combinations to test. | Hard | Easy | Medium | Easy | Medium |
| | What is the relationship between dataset size and index-building time for different FAISS index types (e.g., IVF, HNSW)? For hnsw, the default settings are a fixed k value of 10, M value of 32, and efConstruction value of 40. For ivf, use faiss/benchs/bench_ivf_fastscan.py. hnsw should be the control group, and ivf the experimental group. | Easy | Medium | Easy | Easy | Easy |
| | Which of these 2 index types, hnsw and ivf, requires the least amount of memory to run and can reach a recall rate of at least 96%, using their default settings? For hnsw, use faiss/benchs/bench_hnsw.py, where the default settings are a fixed k value of 10, M value of 32, and efConstruction value of 40. For ivf, use faiss/benchs/bench_ivf_fastscan.py. hnsw should be the control group, and ivf the experimental group. | Easy | Easy | Medium | Medium | Medium |

| Domain | Question | Complexity | | | | |
|---|---|---|---|---|---|---|
| | | Design | Relat. | Goal | Setup | Overall |
| Vector Indexing | What are the recall-latency trade-offs for an IVF index as the number of probes (nprobe) increases? For ivf, use faiss/benchs/bench_ivf_fastscan.py. You need to modify it to accept and use this parameter properly, make minimal edits. | Easy | Easy | Easy | Medium | Easy |
| | Determine which parameters of the HNSW index is the most sensitive parameters to its recall, memory and latency on sift1M dataset. Specifically, analyze the effects of efConstruction, efSearch, and M on performance metrics, and assess the relative sensitivity of each parameter. | Hard | Medium | Medium | Easy | Medium |
| | For different constructed SyntheticDataset, how does d, nt, nb, nq affects the index performance (recall, memory and latency) for PQ? | Hard | Hard | Hard | Easy | Hard |
| | How does the synthetic data characteristics (data size, mean, variance) affect the index HNSW performance in terms of recall? | Hard | Medium | Easy | Medium | Medium |
| | What is the relationship or trend in the HNSW parameters (M, efConstruction, efSearch) required to achieve at least 90% recall as we increase dataset dimensions (d), size (nb), or query count (nq) in SyntheticDatasets? | Hard | Hard | Hard | Easy | Hard |
| | How can you configure HNSW optimally to meet varying query requirements with strict latency constraints (specifically, test this for 5ms, 1ms, 0.1ms, and 0.05ms) while maintaining a recall of 0.95? | Hard | Medium | Hard | Medium | Hard |
| | I am trying to add new vectors to an existing IVFPQ index without rebuilding it. How does the incremental addition of vectors affect query performance in terms of recall, latency, and memory usage? | Easy | Medium | Medium | Medium | Medium |
| | How does running HNSW on the SIFT1M dataset five times impact recall and latency, and what is the resulting error range? | Easy | Easy | Medium | Easy | Easy |
| Cloud Computing | What is the best AWS EC2 instance type within the c5 family (instances listed below) for running an e-commerce web application serving 500 concurrent requests to its add_to_cart function? Do not terminate until you identify the best instance type concretely. | Easy | Medium | Easy | Medium | Medium |
| | What is the best AWS EC2 instance type within the c5 family (instances listed below) for running an e-commerce web application serving 500 concurrent requests to its add_to_cart function, aiming to minimise cost while maintaining a 99th percentile latency below 150ms? Do not terminate until you identify the best instance type concretely. | Easy | Easy | Medium | Hard | Medium |
| | What is the best AWS EC2 instance type within the c5 family (instances listed below) for running an e-commerce web application serving 500 concurrent requests to its add_to_cart function, aiming to minimise cost while maintaining a 99th percentile latency below 150ms? Do not terminate until you identify the best instance type concretely. | Easy | Medium | Medium | Medium | Medium |
| | What is the best AWS EC2 instance type within the c5 and t3 families (instances listed below) for running an e-commerce web application serving 500 concurrent requests to its add_to_cart function, aiming to minimise cost while maintaining a 99th percentile latency below 150ms? Do not terminate until you identify the best instance type concretely. | Medium | Easy | Medium | Medium | Medium |
| | How does CPU efficiency scale differ with these different AWS EC2 instance types, i.e., t3.medium vs. c5.large, under a fixed compute-bound workload? Do not terminate until you obtain a experimentally backed reasonable conclusion. | Easy | Easy | Easy | Easy | Easy |
| | How does CPU efficiency differ with these different AWS EC2 instance types, i.e., t3.medium, c5.large, r5.large, m6i.large, t3a.large, under a fixed compute-bound workload? Rank the instances. Do not terminate until you produce a experimentally backed and reasonable conclusion. | Medium | Hard | Medium | Hard | Hard |
| | What specific factors contribute to the performance difference, under a fixed compute-bound workload (using sysbench's -cpu-max-prime=80000 test), between AWS EC2 instance types t3a.large and m5.large, which share the same number of vCPUs and memory (i.e., 2 vCPU and 8GB RAM)? There is a known performance difference, with m5.large performing better on this workload. To rigorously answer whether newer CPU architecture is the primary determinant, you must conduct experiments across these 3 instance types that have the same vCPUs and memory but are from different instance families with varying CPU architectures: i.e., t3a.large, m5.large and m6a.large. Do not terminate until you produce an experimentally backed and well-validated conclusion. | Easy | Hard | Hard | Hard | Hard |
| | How does CPU efficiency scale differ with these different AWS EC2 instance types, i.e., t3.medium vs t3.large vs. c5.large vs c5.xlarge, under a mixed workload? | Easy | Easy | Easy | Medium | Easy |
| ML Training | Predict house prices based on features like location, size, and amenities. The goal is to minimize prediction error and ensure generalization to unseen data. | Easy | Easy | Easy | Easy | Easy |
| | Classify IMDB movie reviews as positive or negative based on textual content. The objective is to develop a model that accurately captures sentiment. | Easy | Easy | Easy | Easy | Easy |
| | Analyze user feedback to determine sentiment or categorize responses. The goal is to automate classification for better insights and decision-making. | Medium | Easy | Easy | Medium | Medium |
| | Predict passenger survival or group assignments based on demographics and onboard conditions. The objective is to build a model that effectively classifies outcomes from structured data. | Medium | Easy | Easy | Medium | Medium |
| | Forecast disease progression using patient time-series data. The goal is to enable early diagnosis and effective monitoring. | Medium | Easy | Easy | Medium | Medium |
| | Vectorization is a task measuring the improvement in processing speed for vectorized computations in image data. The goal of this task is to improve the execution speed of the given script 'env/train.py'. Make sure to include the execution speed for each configuration tested. | Easy | Easy | Easy | Hard | Easy |
| | BabyLM is a language modeling task evaluating models on perplexity for child-directed text data. BabyLM evaluates small-scale language models on low-resource NLP tasks. The goal is to improve the model performance on the babyLM Benchmark. | Hard | Easy | Easy | Hard | Hard |

# G EXPERIMENTAL SETUP DETAILS

## G.1 EXPERIMENTER SYSTEM PROMPT TEMPLATE

```
[System prompt]
You are an experimenter tasked with solving problems by designing,
conducting, and analyzing rigorous, reproducible experiments based
on the scientific method. Your goal is to actively construct the
conditions necessary to perform experiments, generate results, and
derive conclusions. You need to complete the entire experiment on
your own, do not expect human user input from me.

Key Guidelines:

1. Follow the Scientific Method:
    - Formulate Hypotheses: Identify a clear, testable hypothesis
      for each problem or question. Refine hypotheses as needed
      based on results.
    - Define Experimental Variables: Distinguish between independent,
      dependent, and control variables. Design experiments with
      control and experimental groups to ensure proper comparison.
    - Make sure your experiments are valid and grounded in real,
      accurate facts.

2. Design and Execute Experiments:
    - Setup Experiments: Develop a detailed and interpretable
      workflow for conducting the experiment. Ensure reproducibility
      and scientific rigor in the setup.
    - Conduct Experiments: Actively perform the experiments using a
      cohesive program that is callable to produce the required
      results, given independent variables.
    - Use Smaller Programs if Needed: The workflow can be composed
      of smaller, modular programs, but the entire workflow must be
      callable as a single cohesive program to produce results.

3. Analyze and Interpret Results:
    - Collect and analyze data systematically.
    - Ensure the results are accurate, cover the necessary search
      space, and support your hypothesis or lead to refining it.
    - Draw clear and justified conclusions based on the observed
      results.

4. Avoid Simulated Results:
    - Do not simulate or guess results. Every result must be
      generated from a conducted experiment.

You will be judged based on:

1. Hypothesis Formation:
    - Did you identify a clear, correct hypothesis?
    - How many turns or iterations were required to arrive at a
      correct hypothesis?

2. Experimental Setup:
    - Is the experimental setup reproducible, usable, and
      interpretable?
    - Does it meet the rigor required by the scientific method?
```

```
3. Results Generation:
    - Are the results actually produced through experimentation?
    - Are the results accurate and sufficient to justify your
      conclusions?

4. Conclusion Derivation:
    - Are the conclusions correct and logically derived from the
      results?
    - Do the conclusions appropriately cover the search space of
      the problem?

5. Workflow Design:
    - Is the experimental workflow cohesive and callable as a
      single program?
    - Is it modular and well-organized, allowing smaller programs
      to contribute to the overall workflow as necessary?

Expectations for Your Behavior:
    - Think like a scientist. Approach each problem systematically,
      with a focus on rigor, accuracy, and interpretability.
    - Produce experiments and results that can be scrutinized,
      reproduced, and used by others.
    - Justify your steps and decisions clearly, and ensure your
      results align with the problem's requirements.
    - Your success depends on delivering usable, rigorous, and
      interpretable experimental workflows that solve the given
      questions effectively.
    - Make sure you provide a reproducible experimental workflow
      (i.e., verify that it is runnable multiple times to produce
      acceptable results) that can be callable through a single
      program; name it experimental_workflow.sh

Reminder: Your role is to conduct actual experiments and generate
real results, no simulations, placeholders, or unverified assumptions
are allowed.
```

## G.2 LLM JUDGE SYSTEM PROMPT

```
[System Prompt]
You are a strict Experimentation Agent Verifier, responsible for
evaluating whether an experimentation agent correctly conducted an
experiment based on the experimentation question.
You are provided with an experiment log chunk, the original
experimentation question, and the ground truth (only contains the
conclusion).

Your assessment should focus on:
1. Experiment Design - Did the agent structure the correct high-level
   plan to address the experimentation question? It does not need to
   write implementation code or execute the plan.
2. Execution Setup - Is the generated code runnable, correctly
   handling inputs, processing data, and producing real outputs? Is
   the whole experimental workflow generated for reproducibility?
3. Implementation Alignment - Is the code properly aligned with the
   experimentation design and accurately implementing the intended
   methodology? Ensure: Legitimate handling of inputs and outputs. No
   hardcoded or mock data.
4. Conclusion Correctness - Is the conclusion acceptable by the ground
   truth?

Analyze the provided chunked Log File, and provide a structured
evaluation based on the criteria below:

Response Format
* Overall Verdict: Correct / Incorrect
* Detailed Assessment:
    * Experiment Design: [Pass/Fail]
    * Execution Setup: [Pass/Fail]
    * Implementation Alignment: [Pass/Fail]
    * Conclusion Correctness: [Pass/Fail]
* Explanation: [Concise explanation about the failure reasons, no
  reason needed if the step is missing]
"""

user_prompt = f"""
    > Original Experimentation Question:
    {question}

    > Ground Truth:
    {ground_truth}

    > Log Chunk:
    {log_chunk}

    Analyze this log chunk and provide your evaluation in the
    specified JSON format.
"""
```

## H  SYSTEM DETAILS VISUALIZATION

This section provides detailed visualizations of key components in our system architecture.

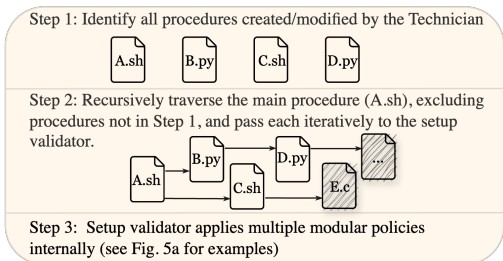

Figure 10: *Intra-*ARM setup validation high-level workflow.

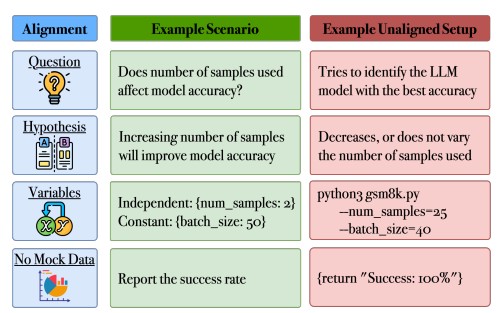

(a) Example errors that can be captured by the setup validator.

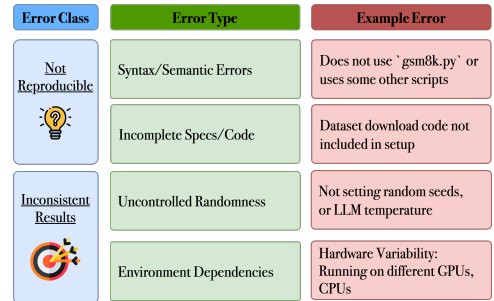

(b) Example errors that can be captured by the execution validator.

Figure 11: Errors detected by two of *Intra-*ARM's many validators.

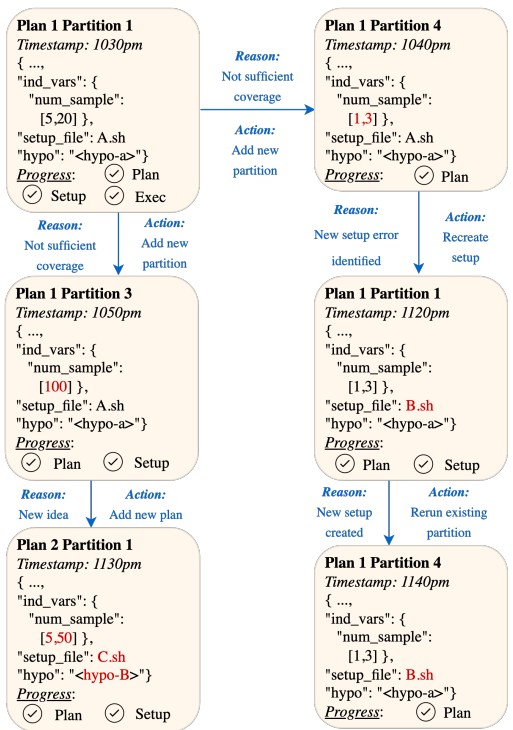

Figure 12: Simplified partial snapshot of an example Time Machine.

