# OpenReview forum: "Curie: Toward Rigorous and Automated Computer Science Experimentation with AI Agents"
_ICLR.cc/2026/Conference — ICLR 2026 Conference Withdrawn Submission_

### Official Review · Reviewer_BmNn · 2025-10-23

**Soundness:** 2
**Presentation:** 2
**Contribution:** 2
**Rating:** 2
**Confidence:** 3

**Summary:**

In this paper, the authors present Curie, an AI agent "framework" (what makes this a framework is a bit unclear) for performing computer science experimentation. One unique feature of Curie is the integration of inter- and intra-agent modules for ensuring experiment and research integrity. There is lots of generic text about "rigor in experimentation" (beginning of `section 2.2`) and "experimental complexity" (section 4.2), which, for me, complicate and obscure the details of what was actually done. My understanding is as follows (following the rather confusing workflow in Figure 3): Curie consists of two conventional LLM-based agents, an architect and technician agent, which produce experiment designs and execute those designs, respectively, and an intermediate "Experimental Rigor Engine" layer that checks the integrity of the produced experiments and artifacts (both at the individual agent level, the intra-agent modules, and at the system level, the inter-agent modules).

Basic details about what these modules are exactly is still rather unclear to me. For example, in Section 3.3, the description of the inter-agent module says that it is responsible for making high-level plans more fine-grained (*how is this operationalized? Are LLMs doing this?*), ensuring that certain control flow is followed (*Where do these control flow rules, such as those shown in the middle of Figure 4, come from?*) and scheduling and managing large-scale expeirments. (It would be very helpful to have, at a minimum, a clear definition of the input and output of these different modules).  In the *Related Work*, the authors claim say that *existing agenets for end-to-end scientific research... rely on ad-hoc prompts to guide predefined workflows..*, which they claim to improve upon, however these integrity modules similarly seem to based on ad-hoc components and predefined workflows.

To test their system, they built a new benchmark. The details of this are left very opaque (e.g., no clear table of examples are provided in the main text, only in Appendix F, outside of the fairly uncompelling running examples in Figures 1-3). The dataset spans the CS domains listed in Table 1 and is based on *real-world influential research papers* (which ones?) and *popular open-source projects* (which projects? these details need to be made more clear in the main text). They compare against two well-known generalist agents, OpenHands and Magentic-one, and use an LLM-as-judge to evaluate different aspects of the agent experiment design and execution process (detailed start on line 402).

On their new benchmark, Curie outperforms both baselines. Intuitively, this shows the value of having experiment integrity checkers, however, the lack of qualitative examples makes it unclear what kinds of exampels their system is able to fix (I would encourage the authors to include such examples, which would likely be more informative than the analysis they report on domain performance and problem complexity performance). Since the concrete details of what these checkers do and how they are operationalized is missing, it is unclear to me what these results say exactly, and what broader insights one can obtain about building generalist LLM agents.

**Strengths:**

-- **A new agentic workflow** (or what they call a "framework") that the integrates agent-level and system-level experiment integrity checking capabilities to reduce problems like hallucination, plan deviation, etc.

-- **A new benchmark** for testing models on computer science experimentation tasks, one built from existing papers.

**Weaknesses:**

-- **Many unclear details about the design of the proposed integrity checking modules** (*see discussion above*). Because of this, it is unclear how others can benefit for their new proposal, or integrate and implement such modules into their own agentic workflows.

-- **Many missing details about their new benchmark** in the main text, e.g., what the examples look like, how it was created, details of whether it is deemed to be of high quality (from what I can see, no human evaluation is reported about its quality, at least in the main text).

-- **Limited empirical validation** of their new system, limited only to their new benchmark. To show general value of their approach, having results on an established benchmark (e.g., some of the benchmarks they discuss in Appendix D) would be very helpful for better assessing the empirical advantage of their approach. It is unclear why such expeirments were not performed.

**Questions:**

-- Is there a particular reason for not evaluating your model on, e.g.,  BixBench, DiscoveryBench, ML-bench?

-- (from above)  Where do the control flow rules for the inter-ARM module (such as those in Figure 4) come from?

-- Are the inter and intra-module implemented via LLM agents? How much engineering effort is put into the partition scheduling component?

---

### Official Review · Reviewer_ePvV · 2025-10-31

**Soundness:** 2
**Presentation:** 2
**Contribution:** 2
**Rating:** 2
**Confidence:** 4

**Summary:**

The paper introduces Curie, and agent scaffold for conducting scientific experiments (in CS) using language models. The authors conducted an evaluation with GPT-4o on a 46-task benchmark they introduced.

AI for science is an important area, and understanding the impacts of LLM-based agents on conducting autonomous (or semi-autonomous) scientific research is critical.

The paper takes some steps towards increasing our understanding of LLM agents' ability to conduct such research. But it suffers from severe shortcomings that prevent me from recommending acceptance. I go into these in the weaknesses section below.

**Strengths:**

- The agent scaffold seems like an interesting contribution. It would be nice to see the effect of adding each of the components to the eventual scaffold through ablations.
- The authors compare against two other prominent agent scaffolds, which Curie outperforms. Were these results able to be replicated (a) on prominent benchmarks that other developers have also tested and optimized their methods for, (b) on test/held-out versions of the benchmarks, (c) across model classes, and if the results (3.4x improvement) still held up, the results would be much more impressive.

**Weaknesses:**

- ***Small number of evaluation samples***: The biggest weakness is the weak evaluation setup. Rather than comparing against existing agent scaffolds on known benchmarks, the authors come up with a new set of 46 questions.
- ***Overclaiming results***: Statements like "Curie represents the first step towards rigorous and automated experimentation" are exaggerated, since there has been such a lot of work in this area over the last few years.
- ***Unclear details on experimental setup***: Statements like "[agents] may be busy handling other partitions" are unclear; couldn't you just respawn additional agents?  What is the purpose of the scheduler if so?
- ***Train/test split?***: Does the experimental benchmark you create come with a train test split? If not, how did you avoid leakage when you were developing the agent scaffold? Also, what did the iterative process for coming up with the scaffold look like? Did you create it in one shot (If so, why do you think this is the best scaffold, since there could be other improvements you come up with?) If not, do you have ablations of the things you tried while developing the scaffold?
- ***Only one model tested***: Agent scaffolds are highly sensitive to the choice of model. The paper's main contribution is an agent scaffold, so just testing it on one model (4o) doesn't offer a complete picture of how well the model would perform.
- ***No details about how well the LLM judge performs***: The authors say they conduct a manual validation of the LLM judge but I couldn't find any details about how well it worked in D.5 or G.2

**Questions:**

I have listed a number of concerns in the sections above. While I think the paper needs fundamental revisions, I would be happy to increase my score if the authors are able to conduct evaluations on standard benchmarks, with a greater number of models, provide more details on the LLM-as-judge manual evaluation, and address my other concerns.

---

### Official Review · Reviewer_ws64 · 2025-11-01

**Soundness:** 3
**Presentation:** 3
**Contribution:** 3
**Rating:** 6
**Confidence:** 3

**Summary:**

This paper introduces Curie, an AI agent framework designed to automate rigorous scientific experimentation in computer science. The system comprises an Architect agent for high-level planning, Technician agents for execution, and an Experimental Rigor Engine with three modules: Intra-Agent Rigor (reliability validation), Inter-Agent Rigor (methodical control), and Experiment Knowledge (interpretability tracking). The authors evaluate Curie on a new benchmark of 46 experimental tasks across four domains (LLM reasoning, vector databases, cloud computing, ML training), achieving 3.4 times improvement over baselines like OpenHands and Magentic-One.

**Strengths:**

- The paper addresses an important and underexplored problem—automating rigorous scientific experimentation rather than just individual coding or reasoning tasks. The focus on enforcing methodical procedures, reliability, and interpretability throughout the experimental process is a meaningful contribution that distinguishes this work from existing agent frameworks.

- The benchmark design is well-motivated and appears comprehensive, with tasks derived from real research papers and open-source projects. The multi-dimensional complexity framework (design, setup, relationship, goal) provides a more nuanced evaluation than simple difficulty levels, and the experiment-centric task formulation better reflects real scientific workflows.

**Weaknesses:**

- The lack of ablation studies is a significant limitation that undermines understanding of which components actually drive performance gains. While the authors argue in Appendix D.4 that ablations are challenging because components are tightly integrated, it seems feasible to at least compare against simpler baselines (e.g., single-agent with validation prompts, or disabling specific rigor policies) to isolate contributions.

**Questions:**

What proportion of the alignment evaluations were manually verified, and what was the agreement rate between the LLM judge and human annotators? It would strengthen the paper to report these statistics and perhaps show examples where manual verification was necessary.

---

### Official Review · Reviewer_SygV · 2025-11-04

**Soundness:** 3
**Presentation:** 3
**Contribution:** 2
**Rating:** 4
**Confidence:** 3

**Summary:**

The paper introduces Curie, a framework for enforcing rigor in using AI agents for automated ML research and experimentation. It decomposes rigor into intra-agent reliability, inter-agent coordination, and experiment interpretability, aiming to improve reproducibility and methodological soundness. Evaluations across diverse tasks from real-world papers show Curie outperforming existing coding agents in consistency and correctness.

**Strengths:**

- The benchmark’s focus on real-world experimental tasks from real papers, albeit a small number of papers that were considered
- The claimed 3.4× improvement is impressive but primarily relative to baselines not built for the same notion of scientific rigor.
- The paper is generally clear and logically structured. Writing is easy to parse and clear.

**Weaknesses:**

- Novelty of the paper. While the contribution of the new benchmark tasks and design philosophy is interesting and seemingly useful, it is unclear to me what new insights this work provides to past work. Several prior frameworks—such as AutoGPT, OpenHands, ScienceAgentBench—already explore automated experimentation with AI agents for ML research. It appears to me that Curie’s novelty lies mainly in its terminology rather than in a fundamentally new design or insights.
- The paper compares two existing coding agents as baselines. These baselines do not have task-specific optimizations for running ML research experiments, so it is unclear how significant the gap in performance is in practice.
- The tasks used for evaluation are from only a very small number of papers (5?). This limits the robustness of the claims the authors are making.

**Questions:**

- In what specific ways does Curie differ mechanistically from prior agent frameworks such as ScienceAgentBench or OpenHands?
- Could you explain how your definition relates to established notions of scientific rigor—such as hypothesis testing, uncertainty quantification, or reproducibility in the statistical sense?
- I would like to see a cost-controlled evaluation of the gains in accuracy [1]

[1] https://arxiv.org/abs/2407.01502

---

### Note · Authors · 2025-11-27

I have read and agree with the venue's withdrawal policy on behalf of myself and my co-authors.